# Natural Products: A Potential Source of Malaria Transmission Blocking Drugs?

**DOI:** 10.3390/ph13090251

**Published:** 2020-09-17

**Authors:** Phanankosi Moyo, Grace Mugumbate, Jacobus N. Eloff, Abraham I. Louw, Vinesh J. Maharaj, Lyn-Marié Birkholtz

**Affiliations:** 1Malaria Parasite Molecular Laboratory, Department of Biochemistry, Genetics and Microbiology, Institute for Sustainable Malaria Control, Faculty of Natural and Agricultural Sciences, University of Pretoria, Private Bag x20, Hatfield, 0028 Pretoria, South Africa; braam.louw@up.ac.za; 2Department of Chemistry, School of Natural Sciences and Mathematics, Chinhoyi University of Technology, Private Bag, 7724 Chinhoyi, Zimbabwe; gmugumbate@cut.ac.zw; 3Phytomedicine Programme, Department of Paraclinical Sciences, Faculty of Veterinary Science, University of Pretoria, Private Bag x04, Onderstepoort 0110 Pretoria, South Africa; kobus.eloff@up.ac.za; 4Department of Chemistry, Faculty of Natural and Agricultural Sciences, University of Pretoria, Private Bag x20, Hatfield, 0028 Pretoria, South Africa; vinesh.maharaj@up.ac.za

**Keywords:** transmission blocking, *Plasmodium*, *Anopheles*, natural products, extracts, malaria, gametocytes, gametes, ookinete, oocyst, endectocide

## Abstract

The ability to block human-to-mosquito and mosquito-to-human transmission of *Plasmodium* parasites is fundamental to accomplish the ambitious goal of malaria elimination. The WHO currently recommends only primaquine as a transmission-blocking drug but its use is severely restricted by toxicity in some populations. New, safe and clinically effective transmission-blocking drugs therefore need to be discovered. While natural products have been extensively investigated for the development of chemotherapeutic antimalarial agents, their potential use as transmission-blocking drugs is comparatively poorly explored. Here, we provide a comprehensive summary of the activities of natural products (and their derivatives) of plant and microbial origins against sexual stages of *Plasmodium* parasites and the *Anopheles* mosquito vector. We identify the prevailing challenges and opportunities and suggest how these can be mitigated and/or exploited in an endeavor to expedite transmission-blocking drug discovery efforts from natural products.

## 1. Introduction

### 1.1. Transmission-Blocking: An Integral Tool for Malaria Elimination

In spite of the many efforts that have been explored to control malaria, the disease still remains a global health threat [1,2,3]. The intricate multistage life cycle of the malaria-causing *Plasmodium* parasite, which spans both development in the human host and mosquito vector, has been one of the major reasons for its survival and continued infection of humans. Each developmental stage of the parasite is characterised by distinct biological processes that causes the variation in stage-specific drug susceptibility [4,5,6,7,8,9]. After hepatic schizogony (liver stage development as initial step after infection with sporozoites transmitted by a feeding female *Anopheles* mosquito), pathology is associated with asexual intra-erythrocytic development of the parasite, typified by progression from ring to trophozoites before schizogony occurs to release daughter merozoites able to infect new erythrocytes and continue proliferation. Sexual development relies on gametocytogenesis of a small fraction of the parasites (~1% of the population) and is characterised by the parasite differentiating through five developmental stages (stages I–V) in the human host to produce mature gametocytes (stage V) able to be transmitted by a feeding mosquito. Once back in the mosquito vector, gamete formation ensues followed by fertilisation and finally oocyst formation before sporogony [10].

For decades, antimalarial drug development efforts have been (rightly so) skewed towards the discovery of chemotherapeutic agents, drugs able to target the symptomatic intra-erythrocytic asexual stage *Plasmodium* parasites and cure a patient of disease and preventing mortality [11,12]. However, this does not eliminate carriage of gametocytes in these patients and indeed, parasite transmission largely continues unabated due to the general inactivity of these drugs against the sexual stages of the *Plasmodium* parasite life cycle. As global malaria programs shift from control to elimination and eradication [13], emphasis has therefore been placed on discovery of additional activities associated with new antimalarial candidates. Not only should such candidates be able to kill asexual parasites and therefore be useful therapeutically, but they should also have transmission-blocking activity, targeting either sexual stages of *Plasmodium* parasites (classified by the Medicines for Malaria Venture as target candidate profile 5, TCP-5, [10,14,15]) or the *Anopheles* mosquito vector (endectocides, TCP-6, [10,16]).

It is anticipated that transmission-blocking drugs will reduce the burden of malaria by substantially decreasing the number of infectious mosquitoes, resulting in significant decline in secondary human infections [14]. In fact, interruption of transmission through vector control targeted interventions, has been at the heart of some of the major success stories in the fight against malaria including elimination of the disease in several countries [17]. However, the efficacy of vector control has plateaued and is undermined by, amongst others, outdoor feeding behaviour of mosquitoes and insecticide resistance [18]. The use of drugs to target the parasite and thereby prevent transmission is therefore an enticing new possibility as add-on to current standard practice. Moreover, the low number of sexual stage parasites marks them for targeting and their non-proliferative nature could decrease the probability of development of resistance to transmission-blocking drugs [19], a fact compromising the use of all antimalarial chemotherapeuticals targeting asexual stages.

Despite these advantages and the growing body of empirical and clinical evidence substantiating its usefulness [20,21,22,23], there is currently only one WHO approved transmission-blocking drug, primaquine. Unfortunately, its use is limited due to toxicity concerns [24] and it cannot be prescribed to pregnant women, breast feeding mothers and infants [25], populations that has a large potential parasite reservoir, which will perpetuate parasite transmission. It is thus imperative to discover new, safe and clinically effective transmission-blocking agents.

### 1.2. Can Natural Products Prove a Panacea for Transmission-Blocking Drug Discovery Efforts?

Natural products are an extensive reservoir of diverse chemical compounds with novel biological targets and mode-of-action (MoA). These qualities have made them a significant component of the global pharmaceutical arsenal with over half of currently commercially available medicinal drugs having been either derived from a natural source or been inspired by natural compounds [26,27]. The malaria field has equally benefitted, with natural products having played a pivotal role in the discovery of chemotherapeutic antimalarial agents with two mainstay malaria chemotherapeutic agents, artemisinin and quinine, both derived from medicinal plants [28,29]. These agents also served as scaffolds for the synthesis of derivatives including artemether, dihydroartemisinin, artesunate, chloroquine and mefloquine. Another antimalarial, atovaquone, also traces its discovery to a plant-derived natural compound [28]. Natural compounds isolated from microorganisms have similarly had a profound impact towards discovery of chemotherapeutic antimalarial agents by providing privileged scaffolds for the synthesis of derivatives including the tetracycline, doxycycline and the lincosamide, clindamycin [30,31].

However, research on natural products as a source of drugs drastically declined towards the end of the 20th century [32]. This was attributed to challenges associated with downstream development of such compounds in medicinal chemistry programmes, particularly due to limited availability of starting material and structural complexity of purified natural product compounds that restricts their synthesis. Further compounding factors include frequent isolation of pan-assay interfering compounds, repeated isolation of known molecules and the non-compatibility of some secondary metabolites with high-throughput screening platforms [32]. Despite this, the emergence of drug resistant microorganisms and limited chemical structural diversity of synthetic libraries has led to a revival of interest in natural products as sources for drug discovery [33]. The recent discovery of structurally unique bacterial-derived antibiotics, teixobactin [34] and darobactin [35] and a anticancer marine alkaloid, trabectedin [36] is fuelling new research. Likewise, malaria research has benefited from this renaissance with the discovery and development of the natural product inspired clinical antimalarial candidates cipargamin [37] and artefenomel [38]. Apart from providing leads, natural products are also opening up new avenues for rational drug discovery efforts through the identification of useful novel biological targets and pathways in *Plasmodium* parasites [39].

Accumulating evidence supports natural products as a source for transmission-blocking drugs targeting the sexual stages of *Plasmodium* parasites and / or the *Anopheles* mosquito. Some natural compounds exhibit a TCP-5 activity profile while others have dual activity with additional potency against asexual parasites (defined with both TCP-1 and TCP-5 activity). The low hit rates of the synthetic compounds against sexual stage *Plasmodium* parasites [40,41,42,43], motivates expansion of the search for transmission-blocking drugs to natural products. This is justified particularly since their diverse chemical space and wide range of pharmacophores could lead to identification of novel lead compounds and associated targets in the parasite and as such avert existing drug resistance challenges. We therefore discuss here the *status quo* of natural products that have been explored for transmission-blocking activity in *Plasmodium* parasites and debate future usefulness of natural products and provide guidance as to standardised strategies to explore this rich source more expeditiously and economically to discover new transmission-blocking hits.

Transmission-blocking screens are typically complex since compounds should show activity primarily against gametocytes in humans, but also has to translate to retained activity against early sporogonic stages (ESS, gametes and/or ookinetes) and oocyst mosquito stages (Figure 1a). Alternatively, compounds active against the *Anopheles* mosquito itself can then be used in the form of endectocides [10,40]. Moreover, the assays used in transmission-blocking screens are technically challenging as they involve multiple biological assay platforms that spans the entire transmission-blocking cascade, with the standard membrane feeding assay (SMFA) serving as the gold standard assay to confirm a block in human-to-mosquito transmission [40]. Until now, transmission-blocking screens for natural products have been largely confined to late-stage gametocytes (stage IV/V gametocytes) whilst screens directly against gametes or oocysts or for identification of endectocides have received the least attention (Figure 1a).

A summary of the current profile of natural compounds that have been screened for transmission-blocking antimalarial potential, revealed that 80 pure natural product compounds (and 11 derivatives generated from some of these pure compounds) have been investigated for some form of transmission-blocking activity. Of this, 21 compounds are from microbial origin and 59 from plants (Figure 1a, Appendix A). In addition to these purified compounds, complex and/or minimal extracts from 37 plant species and 10 herbal products have been associated with at least some transmission-blocking activity. The plant species investigated were drawn from 17 different plant families with Asteraceae, Meliaceae and Combretaceae being the most represented (Figure 1b).

## 2. Effectiveness of Natural Products Against Transmission-Blocking Stages

### 2.1. Microbial-Derived Natural Products 

#### 2.1.1. Ionophores

Ionophores are lipid-soluble carboxylic polyether complexes that facilitate the transportation of ions across cellular membranes [44]. Inspired by the drug repurposing efforts, D’Alessandro et al. [45] screened three ionophores, salinomycin, nigericin and monensin (all originally isolated from different *Streptomyces* sp. [46]), against early- and late-stage *P. falciparum* gametocytes in vitro.

All three compounds were highly active (IC_50_ < 200 nM, Figure 2, Table 1) against both gametocyte stages, with salinomycin showing preference to late-stage gametocytes [45]. The ionophores were able to inhibit development of *P. berghei* gametocytes into early sporogonic stages (ESS) in vitro and the transmission-blocking properties of these compounds was confirmed in vivo using the standard membrane feeding assay (SMFA) (Table 1) [45]. Maduramicin, an ionophore produced by the actinomycete *Actinomadura rubra*, [47] has transmission-blocking properties both in vitro and in vivo [48,49], killing late-stage *P. falciparum* gametocytes (IC_50_ < 200 nM) (Figure 2, Table 1). This ionophore is fast acting, reducing late-stage gametocyte viability by >90% 12 h post treatment, with morphological changes evident even 1 h after drug exposure. This is similarly reflected in in vivo transmission-blocking activity where oocyst development was significantly blocked by maduramicin following exposure of gametocytes to drug for only 90 min prior to mosquito feed [48].

#### 2.1.2. Peptides, Glycosides and Miscellaneous

The proteasome inhibitor, epoxomicin, is one of the most widely investigated peptides routinely used for transmission-blocking as a reference drug for in vitro gametocytocidal assays [48,50,51,52,53,54]. It has potent (IC_50_ < 10 nM, Figure 2, Table 1) in vitro activity against late-stage *P. falciparum* gametocytes [50,51,53,54], with sex-specific preference towards *P. falciparum* micro-gametes in vitro [50,55]. In vivo, epoxomicin completely blocks the formation of *P. falciparum* oocysts in *An. stephensi* [55]. The peptide carmaphycin B targets the β5 subunit of the yeast 20s proteasome, a well characterised antimalarial target [56]. Carmaphycin B is potent (IC_50_ < 1 µM) against both intra-erythrocytic asexual *P. falciparum* parasites and gametocytes, with 40-fold preference towards asexual parasites (Figure 2, Table 1) [57]. Toxicity concerns with this compound resulted in norleucine replacement of the methionine moiety and racemic changes on valine, drastically improving selectivity of a new derivative [57].

Cyclic oligopeptides have been explored including the antibiotic thiostrepton, which is moderately active against intra-erythrocytic asexual *P. falciparum* parasites (IC_50_ = 8.9 µM) with a dual MoA: blocking protein translation in the apicoplast and inhibiting the 20s proteasome of the parasite [58]. Thiostrepton is similarly only moderately potent against the five development stages of gametocytes (IC_50_ ranging from 1.82 to 3.4 µM) [4], but has a 14-fold enhanced activity against micro-gametes compared to macro-gametes (Figure 2, Table 1) [9]. This compound significantly reduces *P. berghei* oocyst development in *An. stephensi* mosquito midguts as well as reducing the number of sporozoites per mosquito [59]. Dactinomycin (a known transcription inhibitor in eukaryotic cells) and romidepsin (histone deacetylase inhibitor) [49] both show sub-micromolar gametocytocidal activity (Figure 2, Table 1), with in vivo transmission-blocking activity only confirmed for romidepsin [60]. Although these oligopeptides do show potency, their large MW and poor solubility detracts from their development as TCP-5 candidates.

The glycosides adriamycin (a DNA synthesis inhibitor) and plicamycin (a RNA synthesis inhibitor) [49] similarly show sub-micromolar gametocytocidal activity (Figure 2, Table 1), indicating that inhibitors of essential nucleotide synthesis processes are affective against the transmissible forms of the parasite. This extends to transcription inhibitors such as puromycin [4,6,7], with equipotent in vitro activity against all five development stages of *P. falciparum* gametocytes [4]. This compound additionally has the advantage of being fast acting against *P. falciparum* macro-gametes (< 1 h) [50]. Similarly, the antibiotic cycloheximide has an almost exactly similar profile to puromycin, killing all *P. falciparum* gametocyte stages, and being fast acting against macro-gametes [50], whilst also blocking *P. berghei* ookinetes development (Figure 2, Table 1) [61].

The macrolide chlorotonil A is highly potent against late-stage gametocytes (Figure 2, Table 1) [62]. Despite a plethora of investigations examining their transmission-blocking potential, the antibiotics tetracycline, fosmidomycin and deferoxamine have consistently proved to be inactive against both *P. falciparum* gametocytes (IC_50_ values >12.5 µM) and macro-gametes with the latter two compounds additionally unable to block *P. berghei* ookinete development in vitro [4,50,61]. All three compounds failed to significantly reduce the development of *P. falciparum* parasites into oocysts in mosquito vector (Appendix A) [63].

Preliminary assessment of two usnic acid derivatives, designated BT37 and BT122, showed them to be potent in vivo (both had > 99% inhibition of oocyst formation at 250 µg/mL) [64]. Dose-response studies for inhibition of oocyst formation were estimated using logistic regression to range from 35 to 234 µM for both derivatives [64]. While both these derivatives were incapable of blocking exflagellation of mature micro-gametes, they did inhibit transformation of zygotes-ookinetes (Appendix A) [64].

#### 2.1.3. Mycotoxins

Fibrinogen-related protein 1 (FREP 1) is one of the many proteins that facilitate mosquito infection by *Plasmodium* parasites and thus transmission [65]. An in vitro screen of a library of crude fungal extracts for compounds that disrupt interaction of FREP 1 with *Plasmodium* parasites identified three active extracts, with that from *Aspergillus niger* (92% inhibition of FREP 1-*Plasmodium* association) being the most potent [66]. *P*-orlandin was identified as the active principle from this extract and has in vivo transmission-blocking activity against oocysts (Figure 2, Table 1) [66]. Aphidicolin (a DNA synthesis inhibitor, mycotoxin from *Cephalosporum aphidicola*) [67,68] is active by inhibiting exflagellation of *P. falciparum* micro-gametes [64], without displaying overt toxicity (Figure 2, Table 1).

**Table 1 pharmaceuticals-13-00251-t001:** Transmission-blocking activity of microbial-derived natural product compounds. (Further details provided in Appendix A).

Compound	MW	cLogP	Transmission-Blocking Stage Activity (IC_50_, µM/% inhibition @ >5 µM *^a^* or <0.5 µM *^b^*)						References
			EG	LG	Mic	Mac	ESS	Ooc	
**Ionophores**									
Salinomycin	751	5	0.014	0.006			0.035	0.002 *^c^*; 0.018 *^d^*	[45]
Nigericin	724	4.69	0.003	0.001					[45]
Monensin	670	3.74	0.002	0.006			0.017	0.002 *^c^*; 0.001 *^d^*	[45]
Maduramicin	934	1.47		0.015				100% *^e^*	[48,49]
**Peptides, glycosides and miscellaneous**									
Epoxomicin	554	2.12	99.8% *^a^*	0.0004	Inactive	0.008		100% *^b^*	[48,50,51,52,53,54]
Carmaphycin B	515	3.31		0.160					[57]
Thiostrepton	1664	−1.04	2.8	1.8	0.096	1.4	8	Active *^a^*	[4,9,59]
Dactinomycin	1255	0.6		0.015					[49]
Romidepsin	540	1.39		0.637				Active *^b^*	[49,60]
Adriamycin	579	0.36		0.526					[49]
Plicamycin	1085	0.25		0.833					[49]
Puromycin	471	−0.22	0.103	0.110		100% *^a^*			[4,6,7,50]
Cycloheximide	281	1.3	0.6	0.477		100% *^a^*	100% *^a^*		[50,61]
Chlorotonil A	479	4.81		0.030					[62]
**Mycotoxins**									
*P*-Orlandin	410	3.18						56.7% *^a^*; 35.3% *^a^*	[66]
Aphidicolin	338	2.39			100% *^b^*				[64]

*^a^* % inhibition > 5 µM, *^b^* % inhibition < 0.5 µM, *^c^* Oocysts intensity; *^d^* Oocysts prevalence; *^e^* % inhibition of oocysts intensity at 4 mg/kg. MW and consensus LogP (cLogP) calculated using SwissADME online suite [69]. EG–early-stage gametocytes; LG–late-stage gametocytes; Mic–micro-gametes; Mac–macro-gametes; ESS–early sporogonic stages; Ooc–oocysts.

### 2.2. Plant-Derived Natural Products 

#### 2.2.1. Terpenes and Terpenoids

Sesquiterpene lactones are emerging as a good starting point in search of transmission-blocking drugs with several studies proving their potential. The most widely investigated member of this class of compounds is artemisinin, along with its derivatives (Figure 3, Table 2). These agents have consistently been shown in vitro to be potent against early-stage *P. falciparum* gametocytes [4,5,7] but their activity against late-stage gametocytes is rather ambiguous with conflicting data (in some instances >100-fold differences in IC_50_ values) [4,6,7,51,70]. Such discrepancies can be explained by variation in stage composition of parasite cultures and dissimilarities in sensitivity of the different assay platforms used [40]. Nonetheless, artemisinin sterilises mature micro-gametocytes and blocks macro-gamete development [9,50]. While artemisinin does not inhibit *P. berghei* ookinetes development in vitro, it blocks *P. falciparum* oocysts formation in *Anopheles* mosquito [63,71]. Clinical studies have shown artemisinin derivatives to reduce gametocyte density and carriage time [72,73]. However, artemisinin-based combination therapies (ACTs) are unable to clear off the transmittable mature stage V gametocytes clinically [74].

Additional sesquiterpene compounds have been investigated from plant species belonging to the Asteraceae family. Parthenin and parthenolide (from the Asteraceae family members *Parthenium hysterophorus* and *Tanacetum parthenium*, respectively), inhibit exflagellation of micro-gametes and block ookinete-oocysts development [75]. From another Asteraceae plant species, *Artemisia afra*, two previously undocumented gametocytocidal guaianolide sesquiterpene lactone compounds (1α,4α-dihydroxybishopsolicepolide and yomogiartemin) were shown to have µM gametocytocidal activity, the former with a three-fold selectivity towards late-stage compared to early-stage gametocytes (Figure 3, Table 2, Appendix A) [76]. From *Vernonia amygdalina* (Asteraceae), two sesquiterpene lactones were isolated, vernodalol and vernolide, with both showing only marginal ESS activity (Figure 3, Table 2, Appendix A) [77]. A germacranolide sesquiterpene lactone from *Daucus virgatus* (Apiaceae), daucovirgolides G, was the only compound with marked potency in vitro, strongly inhibiting ESS development (Figure 3, Table 2, Appendix A) [78,79].

Taxol (a diterpene isolated from the plant *Taxus brevifolia* (Taxaceae) [80]) (Figure 3, Table 2, Appendix A), that inhibit transformation of *P. gallinaceum* zygotes into ookinetes in vitro by targeting microtubules, within 6 h [81]. Furthermore, zygotes exposed to different dosages of these drugs for 4 h failed to develop into oocysts in midguts of *Aedes aegypti* mosquitoes [81].

*Azadirachta indica* (Meliaceae), the neem tree native to India where it has been used for >3500 years for malaria treatment, has been comprehensively studied for its transmission-blocking activity [82,83,84,85,86,87,88,89]. *A. indica* fractions are active in vitro against both early- and late-stage *P. falciparum* gametocytes (IC_50_ = 0.001 µg/mL) [85,87]. Transmission-blocking activity of *A. indica* has been conclusively demonstrated in vivo (by blocking *P. berghei* gametocyte-ESS development) and ex vivo (inhibiting *P. falciparum* gametocyte and oocyst development) [83,84]. The potency of *A. indica* against sexual stages of *Plasmodium* has been ascribed to limonoids (a class of terpenoids produced by the plant species), with azadirachtin A being the most prominent. Azadirachtin A, along with three of its synthetic derivatives (Appendix A), are similarly potent against *P. berghei* micro-gametes (IC_50_ ranging from 1.8 to 2.7 µM) (Figure 3, Table 2, Appendix A) [82]. Structure-activity relationship analysis showed that the hemi-acetal moiety on carbon-11 to be critical for the observed pharmacological effect of these compounds. Azadirachtin A additionally inhibits exflagellation of gametes ex vivo and blocks development of ESS [88]. The MoA of azadirachtin A has been elucidated to be an impairment of microtubules formation during exflagellation [90]. In contrast to its potency against gametes, azadirachtin A is inactive against asexual *Plasmodium* parasites [90]. Another limonoid shown to have activity against ESS stages is deacetylnimbin [89]. Unlike azadirachtin A, deacetylnimbin has the advantage of being thermally and chemically stable [89]. This is an important property to consider in developing drugs targeting the ESS development process in the mosquito (contraceptive drugs) since they ought to have a long half-life equal to the peripheral circulation period of gametocytes which can be as long as 55 days [89,91]. Structural comparisons between deacetylnimbin, azadirachtin A and other *A. indica* compounds namely, nimbin (poorly active against ESS) and salannin (inactive against ESS), suggested that the presence of a free hydroxyl moiety was crucial for potency [89]. Gedunin, a limonoid highly active in vitro against asexual *Plasmodium* parasites, has been inferred to have an inhibitory effect on the development of oocysts in the mosquito vector while existing data suggests both azadirone and azadidarione to be incapable of blocking oocysts development (Appendix A) [84].

#### 2.2.2. Alkaloids, Steroids and Miscellaneous

The gametocytocidal activity of quinine has been a subject of investigation since the 1940s [92]. In some studies, quinine is reported to be more selectively potent towards early-stage *P. falciparum* gametocytes than to late-stage gametocytes (> 15-fold variation in IC_50_ values) (Figure 3, Table 2, Appendix A) [4,5,51], whilst other studies show a six-fold late-stage gametocytes preference [7,49]. Interestingly, quinine is reported to be active against *P. vivax* and *P. malariae* gametocytes [93]. While it has poor activity inhibiting development of *P. falciparum* macro-gametes [50] and is incapable of arresting *P. berghei* ookinete development in vitro [61], quinine is able to block *P. falciparum* oocyst development in vivo [63]. Other alkaloids including dihydronitidine and heitziquinone, isolated from the plant species *Zanthoxylum heitzii* (Rutaceae), also showed activity against ookinete development in vitro [94]. The quinazoline alkaloid, tryptanthrin and its synthetic derivatives designated NT1 and T8, have significant gametocytocidal activity in vitro (Figure 3, Table 2, Appendix A) [95]. However, of the three agents only NT1 strongly inhibited exflagellation of micro-gametes (Appendix A) [95]. Cryptolepine and a root extract of its parent plant, *Cryptolepis sanguinolenta* (Lindl.) Schlechter (Periplocaceae) both demonstrate moderate gametocytocidal activity [96]. The MoA of cryptolepine on asexual *Plasmodium* parasites has been deciphered to be partly due to inhibition of β-haematin formation [97], a non-viable late-stage gametocyte target [4]. Another alkaloid with demonstrated late-stage gametocyte activity is the protein translation inhibitor omacetaxine [49].

While steroids have this far received minimal attention within the malaria transmission-blocking drug discovery field, a few studies have provided interesting insights into their TCP-5 credentials. Withaferin A (a transcription inhibitor) is one such compound being highly potent against late-stage gametocytes (Figure 3, Table 2) [49]. Three steroids, designated SN-1, SN-2 and SN-4, isolated from the plant, *Solanum nudum* Dunal (Solanaceae), were assessed against ex vivo *P. vivax* parasites. Only compounds SN-1 and SN-2 significantly reduced infectivity [98], although this could not be discerned from solubilising agents included such as polyvinylpyrrolidone (PVP). Encouragingly, SN-2 further significantly reduced oocyst density, a phenotypical effect not observed for PVP (Appendix A) [98]. The results from this study are important as they do point to natural products being useful in targeting sporogonic stages of *P. vivax*. In a recent study, a derivative of the steroid sarachine, designated 1o, was demonstrated to be active against early, mid and late-stage *P. falciparum* gametocytes as well as in blocking *P. berghei* oocysts development in vivo (Appendix A) [99].

Additional screens for transmission-blocking activity associated with plant extracts include a screen of extracts from 12 plant species against late-stage gametocytes of *P. falciparum* in vitro with only extracts of five species, *Terminalia macroptera* (Combretaceae), *Combretum collinum* (Combretaceae), *Argenome mexicana* (Papaveraceae), *Zanthoxylum zanthoxyloïdes* (Rutaceae) and *Lophira lanceolate* (Ochnaceae) [100]. Most extracts had moderate activity (IC_50_ ranging from 20.6 to 54.7 µg/mL) with only the stem bark ethanol extract of *L. lanceolate* demonstrating good activity (IC_50_ = 11.4 µg/mL) [100]. A bioassay-guided approach led to the isolation of seven biflavonoid compounds from *L. lanceolate* including lophirone E which was 100-fold more active towards late-stage gametocytes compared to asexual stage *P. falciparum* parasites (Figure 3, Table 2) [101]. Interestingly, screening *L. lanceolate* extracts against ESS led the isolation of a different set of compounds, (glucolophirone C, and the lanceolins A and B, IC_50_ values ranging from 10.95 µM to 113.58 µM), indicating stage-specific activities (Appendix A) [102].

Paton et al. [103] recently demonstrated that exposure of female *Anopheles* mosquitoes to relatively low concentrations of atovaquone (an analogue of a plant-derived natural compound which targets cytochrome *b*) shortly after *P. falciparum* infection rapidly blocked zygote-ookinete development inside the mosquitoes midgut. This consequently led to failure of oocysts development, rendering the mosquitoes non-infective (Appendix A) [103]. Atovaquone could be administered in a way that mimicked contact with an insecticide on a bed net. Its lipophilic nature allowed for its rapid absorption via the mosquito’s legs and into the midgut where it exerted its sporogonic effect. The study opens up new, previously unexplored avenues which, if properly exploited, may have a profound effect on malaria transmission contributing immensely towards the elimination and eradication solution.

**Table 2 pharmaceuticals-13-00251-t002:** Transmission-blocking activity of plant-derived natural product compounds. (Further details provided in Appendix A).

Compound	MW	cLogP	Transmission-Blocking Stage Activity (IC_50_, µM/% inhibition @ > 5 µM *^a^*)						References
			EG	LG	Mic	Mac	ESS	Ooc	
**Terpenes and Terpenoids**									
Artemisinin	282	2.5	0.012	0.037	0.224	0.120	Inactive	93% *^a^*	[4,6,7,9,50,51,63,70,71]
1α,4α-*	320	0.97	17.5	6.3					[76]
Vernodalol	392	1.45					18.7		[77]
Daucovirgolide G	446	3.63					82.3 *^b^*; 48.4 *^c^*		[78,79]
Taxol	853	3.39					~80% *^a^*		[81]
Azadirachtin A	720	1.08			3.5		17.2		[82,88]
Deacetylnimbin	498	2.77					6 to 25		[89]
**Alkaloids, Steroids and Miscellaneous**									
Quinine	324	2.81	0.44	0.318		29% *^a^*	22.6%*^a^*	85%*^a^*	[4,5,7,49,50,51,61,63]
Dihydronitidine	349	3.65					1.7		[94]
Tryptanthrin	248	2.16	95% *^d^*		Inactive				[95]
Omacetaxine	545	2.47		0.083					[49]
Withaferin A	470	3.45		0.372					[49]
Lophirone E	372	3.95		0.14					[101]

*^a^* % inhibition at 5 µM; *^b^* ESS development; *^c^* Zygote-ookinete development, *^d^* % Inhibition of development at concentration equal to IC_90_ value against asexual stages; * 1α,4α-dihydroxybishopsolicepolide. MW and cLogP calculated using SwissADME online suite [69]. EG–early-stage gametocytes; LG–late-stage gametocytes; Mic–micro-gametes; Mac–macro-gametes; ESS–early sporogonic stages; Ooc–oocysts.

### 2.3. Herbal Remedies as Gametocytocidal Agents

Ten herbal products used for malaria treatment in Ghana showed activity (at 100 µg/mL) in vitro against both early and late-stage *P. falciparum* gametocytes. Interestingly, at 1 µg/mL, the herbal product YF, was significantly more potent against late-stage gametocytes in comparison to early-stage gametocytes, whereas herbal product RT used at sub-optimal concentrations (IC_10_ of asexual parasite stages) had the lowest number of gametocytes [104], indicating some preferential killing of gametocytes in these extracts. Some asexual *Plasmodium* parasite cultures treated with herbal products also had a higher gametocytaemia in comparison to untreated cultures [104], implying that this form of stress (similar to that observed for some antimalarial drugs [105,106,107,108]) induces transformation to sexual development. This makes it that much more important to identify compounds that do kill early- and late-stage gametocytes.

### 2.4. Endectocidal Activity of Plant Extracts Against Anopheles

While extracts of plants and plant-derived natural products have been investigated primarily as insecticides or larvacides (reviewed by Rongnoparut et al. [109] and Kishore et al. [110]), their endectocidal activity (where mosquitoes ingest either the extracts of plants or plant-derived compounds and is thereby killed) is poorly explored. A model endectocidal drug for transmission-blocking is ivermectin [111], a 16-membered macrocyclic lactone semisynthetic derivative drug of avermectin, which is a complex natural product originally isolated from the bacterium *Streptomyces avermectinius* [112,113]. Its MoA is associated with hyperpolarization of cells due to influx of Cl^−^, due to inhibition of glutamate-gated chloride channels (GluCl) [114,115]. In a clinical study, ivermectin decreased *An. gambiae* and *An. funestus* mortality by four to seven-fold, 24 h after ingestion [116], and kills outdoor-feeding *An. arabiensis* mosquito vector when delivered through cattle [117] for up to 21 days post-treatment [118]. Enticingly, ivermectin also possesses good activity against asexual *P. falciparum* parasites and late-stage gametocytes (IC_50′_s of 0.1 and 0.5 µM, respectively, Appendix A) [119]. It remains to be seen if this will translate to epidemiological impact in decreasing the parasite transmission burden, in addition to its success against the mosquito vector.

In the search for new endectocides, Kenyan plant species (including *Tithonia diversifolia* (Asteraceae) and *Ricinus communis* (Euphorbiaceae)) were active at LC_50_ values of 8.30 and 8.69 mg/mL after 3 days and 1.53 and 2.56 mg/mL after 7 days of feeding, respectively [120], with two active compounds isolated (3-carboxy-4-methoxy-*N*-methyl-2-pyridone and ricinine). Interestingly, the survival of mosquitoes fed on fruits of *Mangifera indica* (Anacardiaceae) or parts of *Thevetia neriifolia* (Apocynaceae) or *Barleria lupilina* (Acanthaceae) was decreased by 50–95% [121]. *A. indica* and *Z. heitzii* extracts also have pronounced endectocitocidal properties [122,123].

### 2.5. Transmission-Blocking Activities of Synthetic Derivatives of Natural Compound Analogues Currently in Clinical Trials

The natural compound analogue and clinical antimalarial drug candidates KAE609 (cipargamin) [124] and OZ439 (artefenomel) (structural design inspired by artemisinin) [125] have transmission-blocking properties both in vitro and in vivo [4,38,126,127]. The spiroindolone KAE609 inhibits in vitro gametocyte development with sub-micromolar concentrations [126], whilst the endoper{oxide OZ439′s potent (IC_50_ < 10 nM) gametocytocidal potency is limited to late-stages in vitro [4]. However, OZ439 inhibits exflagellation (>65% at 10 µM) [71] and macro-gametogenesis (IC_50_ = 0.15 µM) [50]. *P. falciparum* transmission-blocking of both compounds has been confirmed in vivo [50,126]. Although KAE609 cleared *P. vivax* gametocytes [127] within 8 h and OZ439 reduced *P. vivax* gametocytaemia in vivo by >90% within 24 h [38], the clinical efficacy of both compounds against *P. falciparum* gametocytes still remains inconclusive [38,127].

## 3. Future Perspectives

While natural product compounds show varied activity against transmission-blocking stages, pragmatic strategies adopted for further discovery of new entities should be refined to ensure selection of high-quality, potent hits to expedite their subsequent discovery and development. This includes stringent selection of natural compound libraries, plants and microbial species to increase the probability of getting hit compounds, given the expansiveness and abundance of the plant kingdom (>300,000 plant species on Earth [128]). While the ethnobotany approach has been pivotal in the discovery of chemotherapeutic agents [29,129], unfortunately, transmission-blocking is not a topic one comes across in folk medicine, making it difficult to formulate a question that will lead to identification of plants used for this purpose. Having noted that some plant families e.g. Asteraceae and Meliaceae, are rich sources of prolific compounds potent against most transmission-blocking stages (Figure 1b) [75,82,83,84,89], a rational approach in selecting plants for screening against sexual stage *Plasmodium* parasites will have to focus on members of these plant families documented in ethnobotanical surveys. The vast unique marine vegetation that produce novel chemical structures [130] should also be explored for transmission-blocking antimalarials. Novel, previously unculturable bacteria also present huge untapped source of chemical diversity.

Alternative sources are the de novo screen of natural product libraries in medium- to high-throughput format. It is now clearly indicated that driving screens based on asexual stage potency may not identify transmission-blocking specific compounds and as such we recommend parallel screens against different parasite life cycle stages or at the start, screening driven primarily on the transmissible forms, after which activity against other stages can be determined for hits obtained [131,132]. For such natural product libraries, stringent go/no go criteria need to be defined, similar to screening any other small molecule libraries. However, natural product hits need to be clearly evaluated very early on in the screening cascade for drug-like characteristics before proceeding to a hit-to-lead optimisation phase. In line with the innovative thought and approach of the MMV Malaria Boxes [133], we propose the assembling of a box consisting of a set of structurally diverse natural compounds with proven antiplasmodial activity, that could be used as interrogative control set.

Since different assay read-outs might vary with the MoA of small molecules, the reliability of these assay platforms currently used to screen for gametocytocidal activity needs to be rigorously interrogated and ‘standardised’ to screen extracts against sexual stage *Plasmodium* parasites at medium- to high-throughput scale. This will need to extend further to clearly defining protocols to adopt such as reference control compounds (and possible extracts) per each stage to be used per assay (providing bench mark IC_50_ values of each standard reference drug/extract), standard incubation periods and set potency levels (either % inhibition of development/viability at specified concentration or IC_50_ values) that will serve to guide as to which extracts will be prioritised for the next phase, that is, isolation, purification and identification of bioactive principles. All of the above is entirely dependent on the ability to isolate and purify bioactive compounds from extracts. Classical bioassay-guided fractionation approaches have been expansively employed within the malaria field [134,135,136], whilst cutting-edge technologies for improved isolation of bioactive compounds have been developed and should be explored for transmission-blocking discovery [134,135,136,137,138].

There is reason to believe that natural product compounds will retain their potency following oral administration to mosquitoes, such as when it collects a blood meal. On the basis of this knowledge, it is a worth-while effort to screen natural products, reported in literature to be highly potent insecticidal agents, in search of new endectocidal compounds. Another prudent strategy to explore in search of endectocides is to examine natural products known to target unique invertebrate ion channels e.g. GluCl with the advantage of potential increased selectivity towards mosquitos and therefore reduced toxicity. It remains to be seen if such novel compounds will indeed be able to impact epidemiologically to reduce malaria transmission.

## 4. Concluding Remarks

Optimal adoption of a transmission-blocking strategy will be crucial for efforts to eliminate and subsequently eradicate malaria to be successful. Perhaps the clearest evidence of the transmission-blocking role of natural products is that currently the only WHO recommended transmission-blocking drug, primaquine, is a derivative of a natural product compound, quinine. It is therefore encouraging to note that natural products do have a potential as a viable rich source of transmission-blocking drugs. The activity of such products described against sexual stage parasites of the two most prevalent malaria causing species, *P. falciparum* and *P. vivax* and some of the most prolific *Anopheles* vector mosquitoes, therefore encourages the further exploration of the vast untapped natural product resources for malaria elimination strategies.

## Figures and Tables

**Figure 1 pharmaceuticals-13-00251-f001:**
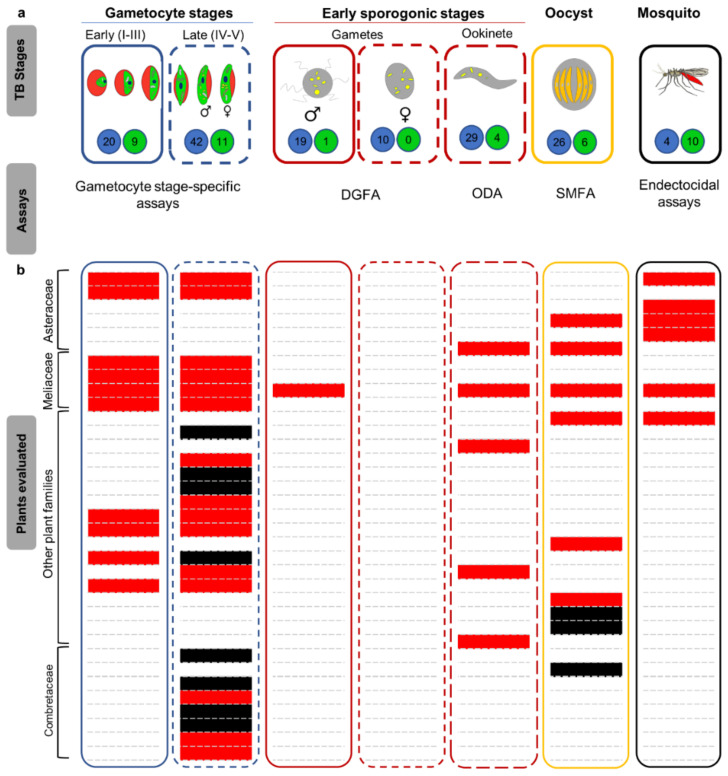
Overview of transmission-blocking assays and natural product origins. (**a**) Sexual stages within the human host are categorised into early-stage (I–III) and late-stage (IV–V) gametocytes. Gametocytes are sexually dimorphic with both male (micro-) and female (macro-) gametocytes found in human host at a ratio of ~1:3.6, respectively. Inside the mosquitoes’ midgut, micro-gametocytes develop into mature micro-gametes, a process called exflagellation. Each micro-gametocyte produces eight micro-gametes while a macro-gametocyte matures into a single macro-gamete. Gametocyte development into gametes is termed gametogenesis. Micro- and macro-gametes fuse together to form a zygote that develops into a motile ookinete. Gamete-zygote-ookinete development constitute early sporogonic stages (ESS). Ookinetes penetrate the midgut wall where they form oocysts which enlarge over time and eventually rupture to release sporozoites. Different assay platforms to assess the activity against different stages include gametocyte stage specific assays (which assess either development, viability, metabolic or redox status), dual gamete formation assays (DGFA) (examine development of mature gametocytes into either micro- or macro-gametes), ookinete development assay (ODA) (examines development of gametes to ookinetes), standard membrane feeding assay (SMFA, assess either the number of oocysts per mosquito (termed oocyst intensity) or total number of mosquitoes with oocysts (termed oocysts prevalence) and endectocidal assays (which examine insecticidal properties of drugs upon ingestion by mosquito). Numbers indicated in blue and green circles indicate number of pure natural compounds and plant extracts screened per each respective stage. TB–transmission-blocking. (**b**) Summary of plant species reviewed for activity against transmission-blocking stages. Quite noticeable is the lack of investigations on gametes. It is also evident that the Asteraceae, Meliaceae and Combretaceae are the most investigated plant families with most species from the latter family being inactive against the respective transmission-blocking stages they were interrogated against. The colour scale indicates active (red) and inactive (black) plants species against specific stages.

**Figure 2 pharmaceuticals-13-00251-f002:**
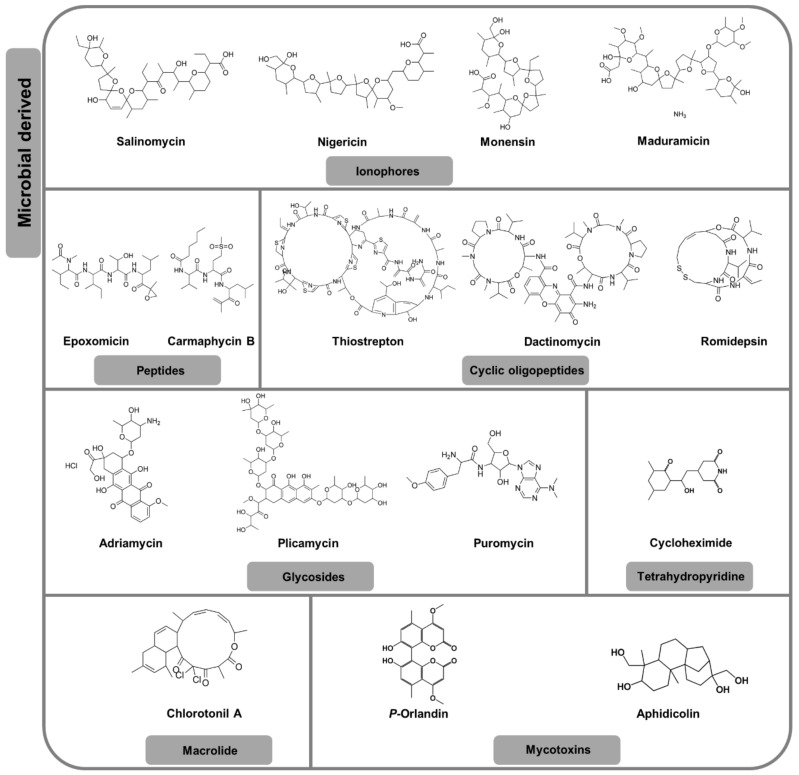
Chemical structures of highly potent microbial-derived compounds targeting *P. falciparum* transmissible stages.

**Figure 3 pharmaceuticals-13-00251-f003:**
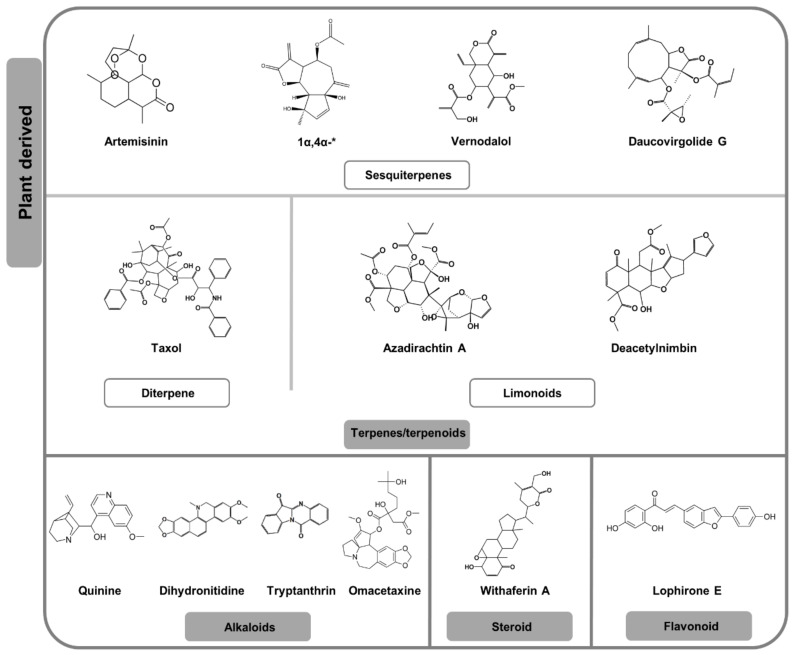
Chemical structures of selected plant-derived compounds with some described activity (including only moderate) against *P. falciparum* transmissible stages. * 1α,4α-dihydroxybishopsolicepolide.

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
