# Peer review of "Natural Products: A Potential Source of Malaria Transmission Blocking Drugs?"

_pharmaceuticals, 2020, doi:10.3390/ph13090251_

Round 1

Reviewer 1 Report

This paper describes efforts to analyse natural products as a potential source of new medicines to block the transmission of malaria. It is well written and clearly laid out and deserves publication in a peer reviewed journal.

The ability to eradicate malaria will ultimately depend on blocking transmission. The only drug in the current portfolio which can be used is primaquine, although safety concerns related to hemolysis in G6PD deficient subjects have limited the dose that can be used to 0.25 mg/kg – and no child friendly forms of the medicine are available for patients <30kg – who represent the majority of the transmitting ‘reservoir’ clinically.

The paper summarises the available data well – this is never easy to do, since the natural products literature does not have common standards of what to report and how to report it.

Figure 2 gives a nice summary of the different existing natural products and their action against the various life cycle stages. The authors wisely chose not to comment in too much detail on compounds with IC50 values greater than 1 uM – knowing that such concentrations are hard to maintain in the plasma safely for most therapeutic agents.

What of course would be interesting would be to relate any of the positive results to the concentrations achievable clinically (or in preclinical species). Again this may be difficult.

In terms of future directions – one thing that I would suggest to add in is the need for standardisation. There is a case to be made for generating a standard set of natural products that can be tested in all of these assays – MMV had a nice collection of small molecule starting points ‘Malaria Box’ which was then used as the basis of many other projects later by various groups. Perhaps this could be a suggestion in the final section?

Author Response

Reviewer comment: What of course would be interesting would be to relate any of the positive results to the concentrations achievable clinically (or in preclinical species). Again this may be difficult.

Response: We certainly agree that this will be interesting to have however, as the reviewer indicated, it will be a difficult undertaking for this current review as more data from clinical and preclinical studies will be required and is not currently available.

Reviewer comment: In terms of future directions – one thing that I would suggest to add in is the need for standardisation. There is a case to be made for generating a standard set of natural products that can be tested in all of these assays – MMV had a nice collection of small molecule starting points ‘Malaria Box’ which was then used as the basis of many other projects later by various groups. Perhaps this could be a suggestion in the final section?

Response: We would like to sincerely thank the reviewer for this insightful suggestion. We have added the following comment in the 'future perspectives' section in line with this – “In line with the innovative thought and approach of the MMV Malaria Boxes (https://www.mmv.org/mmv-open/malaria-box/about-malaria-box), we propose the assembling of a box consisting of a set of structurally diverse natural compounds with proven antiplasmodial activity, that could be used as interrogative control set”

Reviewer 2 Report

This is a review is based on malaria transmission-blocking substances from natural origin. The manuscript is well written and showed updated and very detailed information about the subject.

General comments:

Please revise line 101: “…have similarly had a profund…”

Line 168: the phrase “for interrogation of activity” could be changed with something such as “to assess the activity against…”, or a similar phrase.

Figure 1: in the line 168 please revise the phrase “transmission-blocking stages”, and do adjustments. The figure starts with the explanation of the sexual stage of Plasmodium in humans, this is not called “transmission-blocking stages”. The term is incorrectly used when referring to the “sexual stages”.

Figure 2: please add in the description of the figure the compound names indicating their derivation as well (from microbial or plant derivation)

Lines 309, 461, 614: check these lines for possible extra space between words

Line 327: place square brackets [] instead of parenthesis for the family name [Asteraceae]

Line 348, 447, 448: place square brackets [] instead of parenthesis before parenthesis (for example in  line 348: [112…])

Line 469: there is an extra full-stop (period)

Author Response

Please revise line 101: “…have similarly had a profund…”

Response: We thank the reviewer for their comments. Change has been made as suggested.

Line 168: the phrase “for interrogation of activity” could be changed with something such as “to assess the activity against…”, or a similar phrase.

Response: Change made as suggested.

Figure 1: in the line 168 please revise the phrase “transmission-blocking stages”, and do adjustments. The figure starts with the explanation of the sexual stage of Plasmodium in humans, this is not called “transmission-blocking stages”. The term is incorrectly used when referring to the “sexual stages”.

Response: Change made as suggested.

Figure 2: please add in the description of the figure the compound names indicating their derivation as well (from microbial or plant derivation)

Response: Change made as suggested, the previous figure 2 has now been split into two for microbial and plant derived, and compound classes indicated. 

Lines 309, 461, 614: check these lines for possible extra space between words

Response: We have taken action as suggested. Thank you.

Line 327: place square brackets [] instead of parenthesis for the family name [Asteraceae]

Response: We appreciate the suggestion from the reviewer, however, we wish to point out this will be inconsistent with style of manuscript published in the Pharmaceuticals journal (e.g. https://www.mdpi.com/1424-8247/13/5/84/htm; https://www.mdpi.com/1424-8247/13/8/175/htm) and so against this background we have chosen to retain the family names in parenthesis.

Line 348, 447, 448: place square brackets [] instead of parenthesis before parenthesis (for example in line 348: [112…])

Response: Following extensive changes made to the manuscript the specific portions referred to by the reviewer have been since been removed.

Line 469: there is an extra full-stop (period)

Response: Change made as suggested.

Reviewer 3 Report

The authors describe a summary of the activities of natural products of plant and microbial origin against the sexual stages of Plasmodium parasites and the Anopheles mosquito vector. Additionally, they try to identify the prevailing challenges and opportunities to expedite transmission-blocking drug discovery efforts from natural products.

This manuscript is not suitable for publication in its current state. Some changes must be made to make this review article more useful to the scientific and academic community (easy to read and didactic)

Here are some comments that I would like to be addressed by the authors:

Major concern:

Review articles are important because they show the state of the art of a topic and also facilitate the understanding of students and researchers towards a particular topic. For this reason, the review articles seek to summarize and explain much information in an easy and didactic way. That is the strong point of a review article, the reason why they exist and are highly cited. In particular, this article has been difficult for me to read, it only has two images and the main table is in supplementary material. I doubt very much that it can serve as a starting point for a student or researcher.

In particular, I think it would have been more practical and educational to organize certain data in tables. Placing a lot of data from IC50 or scientific names of species in the text makes reading the article very heavy. Perhaps including a summary table at the end of each section is the most ideal. It is somewhat ironic that as a researcher the supplementary material is more useful to me than the article itself. This indicates that there is something wrong.

Minor concern:

Introduction section. Reference number 3 is from 2015 (World malaria report 2015), very outdated for the information currently available

Related Figure 1. I recommend the placement of parallel lines or dotted lines in order to better track the colored rectangles

Related Figure 2. From the point of view of drug discovery, it would be very useful to include some physicochemical properties. I recommend including relevant information such as LogP (any method) and molecular weight

Author Response

This manuscript is not suitable for publication in its current state. Some changes must be made to make this review article more useful to the scientific and academic community (easy to read and didactic)

Response: We thank the reviewer for his/her in-depth evaluation of the paper. We do concur that the paper was quite lengthy and some sections did not summarise the main points in a way that it can be easily used as reference text. We therefore did re-evaluate the entire text to make sure it is better organised, and that the text is easier to read and didactic, rather than only a comprehensive summary of literature. We do believe this has made the paper stronger and more useful as reference text to the community. 

Major concern:

Review articles are important because they show the state of the art of a topic and also facilitate the understanding of students and researchers towards a particular topic. For this reason, the review articles seek to summarize and explain much information in an easy and didactic way. That is the strong point of a review article, the reason why they exist and are highly cited. In particular, this article has been difficult for me to read, it only has two images and the main table is in supplementary material. I doubt very much that it can serve as a starting point for a student or researcher.

In particular, I think it would have been more practical and educational to organize certain data in tables. Placing a lot of data from IC50 or scientific names of species in the text makes reading the article very heavy. Perhaps including a summary table at the end of each section is the most ideal. It is somewhat ironic that as a researcher the supplementary material is more useful to me than the article itself. This indicates that there is something wrong.

Response: We appreciate the reviewer's comments and suggestion. To particularly address the point of summarising information in a didactic way, as well as to make data that was available only in the supplementary more available as the reviewer indicated, we have now made the following changes. We made extensive changes to the text to remove excessive references to IC50 values and scientific names (where possible) and rather kept the main essence to make the narrative more logical and easier to read. We then included figures after each section to show the structures of the important compounds referred to. In addition, we then included tables after each section summarising the chemical properties and biological data associated with important compounds. We believe this will indeed make the paper useful as a reference starting point for students or researchers, by being able to understand the impact of the findings from the text whilst simultaneously being able to evaluate compounds structurally and funttionally from the summarised figures and tables, head-to-head. 

Minor concern:

Introduction section. Reference number 3 is from 2015 (World malaria report 2015), very outdated for the information currently available

Response: Change has been made as suggested with WHO 2019 report cited.

Related Figure 1. I recommend the placement of parallel lines or dotted lines in order to better track the colored rectangles

Response: Change has been made as suggested

Related Figure 2. From the point of view of drug discovery, it would be very useful to include some physicochemical properties. I recommend including relevant information such as LogP (any method) and molecular weight

Response: Change has been made as suggested and this information has been incorporated into the new summarising tables in addition to the comprehensive supplementary file.

Reviewer 4 Report

The authors describe a very important point in the evaluation and discussion of antimalarial-activities of natural as well as synthetic small organic molecules - the transmission blocking abilities of these compounds and possible structure-activity relationships.

They have collected, grouped and carefully discussed relevant natural compounds and classes of these compounds with respect to these effects. Especially the description of the different activity at the different stages of parasite infection is crucial and the data presented here is highly relevant and well presented.

What I can say beside strongly recommending the publication of this work is, that - for a medicinal organic chemist - depiction of the relevant basic structures if a whole group of compounds is discussed and specific structures if these compounds are discussed would be very helpful for reading. Thus, I recommend that more structures are shown at important stages of the paper to make clear what it is all about.

Author Response

The authors describe a very important point in the evaluation and discussion of antimalarial-activities of natural as well as synthetic small organic molecules - the transmission blocking abilities of these compounds and possible structure-activity relationships.

They have collected, grouped and carefully discussed relevant natural compounds and classes of these compounds with respect to these effects. Especially the description of the different activity at the different stages of parasite infection is crucial and the data presented here is highly relevant and well presented.

What I can say beside strongly recommending the publication of this work is, that - for a medicinal organic chemist - depiction of the relevant basic structures if a whole group of compounds is discussed and specific structures if these compounds are discussed would be very helpful for reading. Thus, I recommend that more structures are shown at important stages of the paper to make clear what it is all about.

Response: We thank the reviewer for their comments. To address the need raised for more structures, we have now included two figures where the structures of the most interesting/important compounds discussed in the manuscript are shown. All other structures of all other compounds (e.g. derivatives of the important compounds) can also be found in the Supplementary File.